# Fully Canonical Triple-Mode Filter with Source-Load Coupling for 5G Systems

**DOI:** 10.3390/s25010090

**Published:** 2024-12-27

**Authors:** Cristóbal López-Montes, José R. Montejo-Garai

**Affiliations:** Group of Applied Electromagnetics (GEA), Information Processing and Telecommunications Center, Universidad Politécnica de Madrid, 28040 Madrid, Spain; cristobal.lopez@alumnos.upm.es

**Keywords:** 3D filter, bandpass filter, cross-coupling, elliptic response, fully canonical, source–load coupling, transmission zero, triple mode, waveguide filter

## Abstract

This work presents the design of a novel fully canonical triple-mode filter with source–load coupling for 5G applications, exploiting its very compact size for the FR1 band. The design is carried out using circular waveguide technology to attain power handling and low insertion losses. The fully canonical topology allows for increasing the selectivity of the filter since the number of finite transmission zeros is equal to the order of the filter. Given that this topology needs a source–load coupling level that is not possible to achieve with the classical iris ports, coaxial probes are used as input–output ports. A systematic procedure is developed to obtain the initial geometry before the full-wave optimization. The proof of concept is verified by a manufactured prototype at 3.7 GHz with 1.1% relative bandwidth for high coverage of 5G base stations. The results show an excellent agreement between the simulation and the measurement, validating the triple-mode filter and its underlying design process.

## 1. Introduction

Fifth-generation (5G) mobile networks are now a reality as base stations are currently being gradually deployed. Fifth-generation applications are classified into three categories [1]:Enhanced mobile broadband (eMBB) includes use cases where pedestrian experience is improved in terms of capacity, user density or coverage, among other factors.Ultra-reliable and low-latency communications (URLLCs) encompass real-time applications where latency, reliability and availability are critical, such as autonomous vehicles or telemedicine.Massive machine-type communications (mMTCs) cover applications with a high number of devices and low data transmission rates, like the Internet of Things (IoT).

The 5G standard defines two frequency bands: FR1, also known as the sub-6 GHz band, which spans from 450 MHz to 7125 MHz, and FR2, also called the millimeter-wave band, which ranges from 24.25 GHz to 52.6 GHz [2]. The FR1 band shares channels with previous mobile standards, which enables backward compatibility. Moreover, its lower operating frequencies imply a reduction in propagation losses, making it suitable for providing coverage in extensive areas. In contrast, the FR2 band, which operates at significantly higher frequencies, allows for the manufacturing of miniaturized microwave circuits and delivers greater bandwidth at the cost of higher propagation losses. This band is designed for base stations that provide coverage in small areas with high data-rate demand [1].

In this context, the Power Amplifier (PA) is a key element in the radio frequency front-end because it is responsible for ensuring the link coverage and the quality of service (QoS). The requirements of these devices are high efficiency to minimize the power consumption, high linearity due to the sensitivity of multi-carrier modulations to amplitude and phase distortions and a reduced memory effect to manage bit-rates up to 20 gigabits per second [3]. Another important issue is the interference signals generated by the PA, such as intermodulation products, harmonics or spurious signals that decrease the signal-to-inference ratio [4]. The approach to suppress these unwanted signals is to design an output filter for the PA.

Waveguide filters are suitable for this task because of their power handling capability and low insertion losses [5,6,7,8]. Specifically, multimode waveguide filters enable the miniaturization of the device at the expense of increasing the complexity of the topology and the sensitivity of the tuning process. Typically, multimode waveguide filters use geometries that allow for the generation of degenerate modes, such as circular or square waveguides. The proposed triple-mode filter in this work is based on a cylindrical waveguide because of its low insertion losses and high unload quality factor (Q) [5].

Since in the technical literature the list of triple-mode filter designs implemented in different technologies is very extensive ([9,10,11,12,13,14,15,16]), this work only focuses on cylindrical waveguide technology. In [17], the reported filter implements a transmission zero with the coupling among the modes $TE_{113V,H}$ and the mode $TM_{012}$. In [18], a filtering slot antenna is reported based on a triple-mode circular waveguide filter working with the $TE_{111+,-}$ modes and the $TM_{012}$ mode. The couplings between modes are controlled by the geometry and the position of the feeding and radiating slots. In [19], six-pole and seven-pole elliptic filters with two transmission zeros are designed based on an in-line configuration where couplings are implemented by irises. In [20,21], coaxial probes and a metallic post are used to implement two transmission zeros. In [22,23], partially dielectric-loaded cylindrical cavities are used to also implement two transmission zeros. In [22], the couplings are among the degenerate modes $TE_{111x,y}$ and the mode $TM_{01}$, while in [23] the couplings are among the degenerate hybrid modes $HE_{11+,-}$ and the mode $TM_{01}$. In [24], two inner conductors inside the circular cavity allow for the coupling between the degenerate modes $TE_{11A/B}$ and the TEM mode. This configuration generates two transmission zeros and includes tuning screws. Ref. [25] implements the coupling among the degenerate modes $TE_{111s/c}$ and the mode $TM_{010}$ by means of rectangular irises to generate two transmission zeros.

In this work—for the first time to the authors’ knowledge—the design of a fully canonical triple-mode narrow-band filter is presented. In Section 2, the theoretical synthesis is developed along with the selection of the most suitable coupling matrix. In addition, the dimensions of the cylindrical cavity are obtained considering the resonant modes, the Q factor and the spurious-free window. The physical structure of the filter is presented in Section 3 as the pre-design to calculate the initial dimensions of the coupling elements and the input–output coaxial probes. Lastly, the final electromagnetic design and the measurements of the manufactured prototype for experimental validation are presented in Section 4.

## 2. Theoretical Design of the Triple-Mode Filter

The design of the circular waveguide triple-mode filter comprises two closely related processes. The first one is the theoretical synthesis of the transfer function including three transmission zeros to obtain a fully canonical topology. Its response must verify the specifications of both return and insertion losses. The coupling matrix contains all the information to carry out the circuit design.

The second one is the selection of the electromagnetic resonating modes in the cylindrical cavity to obtain the necessary Q factor and the spurious-free window in the lower and upper frequencies. In addition, the Q factor is directly related to the insertion loss. Once the theoretical synthesis is carried out, the next task is to calculate the dimensions of the cylindrical cavity.

### 2.1. Synthesis of the Transfer Function by Means of the Coupling Matrix

The coupling matrix contains all the information regarding the transfer function of the filter considered as a two-port network [26]. By means of the general Chebyshev function, it is possible to obtain the transmission and reflection polynomials for an equiripple response with the desired position of the transmission zeros [27].

Table 1 collects the specifications of the filter. For the synthesis of the low-pass prototype, the required parameters are the order N=3, the return loss level (23 dB) and the position of the transmission zeros in the imaginary axis of the complex plane (real frequencies), $s_{1}=-j9.5,s_{2}=j3.65,s_{3}=j6.27$, following the 5G standard FR1 [1].

Figure 1 shows the topology of the folded fully canonical network (3-3) of the triple-mode filter. The three electrical node resonators will be implemented by three degenerate (same resonant frequency) electromagnetic configurations or modes inside the cylindrical cavity, i.e., the $TE_{112V}$ mode (V stands for vertical polarization), the $TM_{011}$ mode and the $TE_{112H}$ mode (H stands for horizontal polarization).

Following the well-established procedure [27], a systematic extraction of the capacitors, the frequency-invariant reactive (FIR) elements and the impedance/admittance inverters leads to the coupling matrix (Equation 1): $m_{S1}$, $m_{12}$, $m_{13}$ and $m_{s3L}$ are the mainline couplings; $m_{11}$, $m_{22}$ and $m_{33}$ are the self-couplings; and $m_{1L}$ is the symmetric cross-coupling and $m_{SL}$ is the asymmetric cross-coupling.
(1)$$M=\left[\begin{array}{ccccc} 0 & m_{S1} & 0 & 0 & m_{SL}\\ m_{S1} & m_{11} & m_{12} & m_{13} & m_{1L}\\ 0 & m_{12} & m_{22} & m_{23} & 0\\ 0 & m_{13} & m_{23} & m_{33} & m_{3L}\\ m_{SL} & m_{1L} & 0 & m_{3L} & 0 \end{array}\right]$$

With the purpose of handling a wide range of possibilities in the values of these couplings, mainly in the case of the critical source–load coupling, three different coupling matrices are synthesized and listed in Table 2. All of these matrices result in the same response shown in Figure 2. However, Matrix 1 has the minimum value for the asymmetric cross-coupling $m_{SL}$ implemented by coaxial probes as will be discussed later. For this reason, Matrix 1 is chosen for the design of the triple-mode filter.

### 2.2. Cylindrical Cavity Design: Radius, Length, Q Factor and Spurious-Free Window

It is well known that a metallic cavity can support an infinite number of electromagnetic field configurations or modes [28]. Suitable dimensions of the cavity allow for operating several independent resonant modes at the same frequency (degenerate modes). Thus, a dramatic saving of space is possible using several resonant modes in the same physical structure.

The graphic representation of the resonant frequencies using the mode chart eases finding the points of the triple-mode filter design. In the case of the circular waveguide, all the modes are degenerate since two families coexist (V and H polarization), except for those modes with no angular variation (first subscript equal to zero). The selection of the triple-mode intercept point is set by the spurious-free window and the Q factor, which determines the insertion loss level.

Figure 3a shows the selected point, i.e., the intersection of the $TE_{112V}$, $TE_{112H}$ and $TM_{011}$ modes. This point has an abscissa value ${\left(2a/d\right)}^{2}$ = 0.323 and ordinate value ${\left(2af\right)}^{2}$ = 600.23 that corresponds to the dimensions radius a = 33.11 mm and length d = 116.41 mm for the resonant frequency f = 3.7 GHz. The vertical dashed line shows the free spurious window limited by the $TM_{010}$ and $TM_{012}$ modes. Figure 3b shows this window between the resonant frequency of the lower spurious mode $TM_{010}$ = 3.47 GHz and the resonant frequency of the upper spurious mode $TM_{012}$ = 4.32 GHz, in total 850 MHz. In this manner, both the operation band and the transmission zeros are inside the window.

Figure 4 shows the Q factor of the three modes under analysis versus the aspect ratio of the cylindrical cavity. The frequency is 3.7 GHz and aluminum metallic walls (electric conductivity $\sigma_{Al}$ = 35.4 MS/m) are considered. The analytical Q factor of $TE_{112V,H}$ modes is 19,655 and for the $TM_{011}$ mode it is 15,168. Both values are ideal in the sense that the manufacturing process and the roughness of the internal walls are not taken into account.

## 3. Electromagnetic Pre-Design of the Coupling and Tuning Elements

Once the theoretical synthesis is carried out and the dimensions of the cylindrical cavity are calculated, the next step comprises the electromagnetic design of the couplings between the resonant modes, the source and the load, according to the network topology shown in Figure 1 and the values of the coupling Matrix 1 in Table 2.

At this point, it must be noted that the electrical response of the filter is asymmetric, with nonzero self-couplings $m_{11}$, $m_{22}$ and $m_{33}$. Therefore, the filter is asynchronously tuned and, consequently, it is mandatory to ensure the tuning of the three modes independently. In addition, the couplings between modes must be carried out without generating other unwanted couplings.

Figure 5 shows the magnitude of the transversal component of the electric field for the modes $TE_{112V,H}$ and $TM_{011}$ versus the length of the cylindrical cavity normalized to unity, the position of the tuning and coupling screws and the input–output coaxial probes. There are three screws to achieve the intermode couplings: the $m_{12}$ coupling is implemented by the vertical screw coupling $TE_{112V}$ and $TM_{011}$ modes; the $m_{23}$ coupling is implemented by the horizontal screw coupling $TM_{011}$ and $TE_{112H}$ modes; and the $m_{13}$ cross-coupling is implemented by the screw in the $45^{\circ }$ coupling $TE_{112V}$ and $TE_{112H}$ modes. In addition, there are three tuning screws, one for each mode, two in a vertical position for the $TE_{112V}$ and $TM_{011}$ modes and one in a horizontal position for the $TE_{112H}$ mode. This positioning of the screws minimizes the unwanted interactions. However, it seems clear that in such a sophisticated structure it is impossible to isolate all interactions, even more so considering the accuracy of the manufacturing process. Figure 6 shows the 3D view of the cavity, including the tuning and coupling screws, and the input–output coaxial probes, according to the scheme plotted in Figure 5. At this point, it must be stated that the full-wave electromagnetic simulation will be carried out by means of the *CST Studio Suite* [29].

The input–output coaxial probes are located at the maximum of the transverse electric field of the $TE_{112V}$ and $TE_{112H}$ modes. Since both degenerate modes have two variations along the axial axis of the cavity, the maximums are in the first and third quarters of the cavity (Figure 5). Provided that both modes are rotated $90^{\circ }$, the source probe is placed vertically and the load probe horizontally.

After the physical structure is clearly demonstrated and explained, the next task is aimed at obtaining the initial dimensions of every element that provides the corresponding coupling.

The length of both coaxial probes is calculated by using the method of the group delay of the input reflection coefficient to a single resonator [27,30]. This procedure relates the normalized impedance of the coupling matrix, i.e., the $m_{S1}$ coupling with the group delay. Figure 7a shows the group delay of the input reflection coefficient $s_{11}$ parameter of the resonant cavity illustrated in the inset. As can be observed, two different peak values are obtained since two degenerate modes, i.e., the $TE_{112V}$ and the $TM_{011}$, are resonating. Although the group delay method applies to a single resonator, it is possible to establish a correspondence between every peak and the resonating mode, because the coupling level is inversely related to the group delay level. The lower peak corresponds to the coupling of the $TE_{112V}$ mode. In addition, monitoring the transverse component of the electric field (Figure 7b,c) of both modes at their resonant frequencies, the above statement is confirmed. In this way, the value of the group delay at the lower peak is calculated and therefore the initial length of the probe is obtained. Due to the symmetry of the cavity, the process is similar for the horizontal probe, in this case with the $TE_{112H}$ mode.

The depth of the coupling screws (coupling elements $m_{12}$ and $m_{23}$) is calculated by means of the resonant frequencies of the even and odd mode circuits of two coupled resonators considering the electric and magnetic wall symmetry [27,30]. For solving the eigenmode problem, the coupling level versus the depth of the screw is plotted. Figure 8 shows this graph for the coupling screws $TE_{112H}$-$TM_{011}$ and $TE_{112V}$-$TM_{011}$, which is the same due to the cavity symmetry. In the same way, Figure 9 shows the coupling coefficient ($m_{13}$) versus the depth of the screw for coupling between $TE_{112V}$-$TE_{112H}$ modes, including the 3D model of the cavity and the coupling screw at $45^{\circ }$.

Finally, the depth of the tuning screws must be obtained. The procedure is based on the eigenmode solution of the cavity resonance considering the perturbation introduced by the screw. Since the input–output coaxial probes and the coupling screws affect the resonant frequency, they are included in the electromagnetic model. Figure 10 shows the cavity resonant frequency versus the screw depth of the tuning screw for the $TM_{011}$ mode. In the same way, Figure 11 shows the cavity resonant frequency versus the tuning screw depth for the $TE_{112V}$ mode. As can be observed, the resonant frequency of the $TE_{112V}$ mode is below the center frequency (3.7 GHz) of the filter, making tuning unfeasible. This issue is caused by the effect of the input coaxial probe and the coupling screws that involve the $TE_{112V}$ mode. Due to symmetry, the same issue applies to the $TE_{112H}$ mode. Therefore, the radius and length of the cavity will be included in the final full-wave optimization, adding two more degrees of freedom to cope with this issue.

Table 3 collects the initial values of all the dimensions, i.e., the resonant cavity, the input–output coaxial probes and the coupling and tuning screws. These values lead to the filter response shown in Figure 12. As can be observed, the center frequency is shifted to a lower value, only two reflection zeros are in the bandpass and the third one is maladjusted. In addition, only one transmission zero is visible in the plotted band. This is the starting point for the final full-wave optimization where all the interactions are simultaneously taken into account, in contrast to that accomplished in each of the above steps. It should be noted that to carry out optimization without this initial response, just by *brute-force*, leads inevitably to the failure of this sophisticated design.

In the next section, the final design, the manufacturing process and the tuning procedure in the laboratory are detailed.

## 4. Final Design of the Triple-Mode Filter and Experimental Results

As formerly stated, once the pre-design is finished, the following task is the full-wave optimization of the filter. Next, the prototype is manufactured, tuned and measured in the laboratory to verify the required specifications.

### 4.1. Full-Wave-Optimization of the Final Design

The initial depth values of the coupling and tuning screws as well as the coaxial probes do not encompass the interactions between them as was explained in the previous section. Therefore, full-wave optimization is mandatory with the great advantage provided by the starting point previously obtained. The process begins to center the bandpass with its three reflection zeros, progressively increasing the return loss level. Since the frequency of the transmission zeros is known in advance through the theoretical synthesis, their position and level are gradually controlled. As was said before, the radius and the length of the cylindrical cavity are included as variables to optimize with the aim of increasing the degrees of freedom considering the sensitivity of the fully canonical filter response. The small change in the aspect ratio ${\left(2a/d\right)}^{2}$ from 0.323 to 0.365 makes it possible to achieve the three reflection zeros and the three transmission zeros, maintaining the same spurious-free window. Table 4 collects the final dimensions after full-wave optimization.

### 4.2. Manufacturing, Tuning and Measurement of the Triple-Mode Filter

With the objective of practical validation, a prototype is developed, adjusted and tested in the laboratory. The filter is manufactured in two aluminum pieces, the main body and the cover, using CNC machining. The SMA connectors are screwed directly into the cavity through recesses to ease the hardiness. The internal roughness is carefully considered during the milling. Six precision screws are used to tune and couple the three modes involved in the electrical response. In addition, they are used to compensate for manufacturing tolerances. Lastly, note that the inner part of each screw introduced into the cavity is smooth without thread to reduce insertion losses.

Upon reception of the prototype, the tuning is performed. The fitting begins by introducing the probes and screws to their respective depths according to Table 4. Releasing the cover to have access to the interior, the initial lengths are controlled. This is the starting point for exploiting the combination of the Vector Network Analyzer (VNA) and the *Filter Designer 3D* [29]. This tool offers the possibility of tuning the filter, a challenging task due to the number of tuning elements involved and the extreme sensitivity of the response to small variations in these elements, even more so considering its fully canonical behavior. On the other hand, following our experience, just by observing the scattering parameters on the VNA screen, the insight into the tuning state is unprofitable.

Instead of tuning the scattering parameters, it is much more efficient to extract the coupling matrix measured in real time until the values of Matrix 1 in Table 2 are achieved. By means of error bars, the cited tool shows the difference between the desired and the measured coupling. However, a sequential process must be followed to be successful. Due to its intricate response, the filter is first exclusively tuned in the passband until attaining the three reflection zeros and a return loss level close to the specified value of 23 dB. Subsequently, the tuning is extended to the whole band from 3.5 to 3.9 GHz to achieve the three transmission zeros at their respective frequencies and values. Note that the filter is extremely sensitive to any change in the position of the screws or probes, requiring great care in tuning since all the effects are simultaneously present in the physical cavity. In addition, the manufacturing tolerances must be compensated. It has to do with, in short, an adjustment process with eight elements, two probes and six screws.

Figure 13 and Figure 14 shows the comparison between the simulation and the measurement of the S-parameters once the tuning is finished. As can be observed, the three reflection zeros are in their frequencies, thus achieving the equal-ripple specification of the Table 1. In addition, the positions of the three transmission zeros are very close to those simulated. In summary, the agreement between the theoretical and the experimental results is excellent.

The comparison between the simulation and the measurement of the insertion losses in the passband is shown in Figure 15. The highest value of 0.5 dB is found in the upper limit of the bandpass. In this regard, the narrow relative bandwidth of 1.08 % of the prototype must be highlighted, having a bandwidth of 40 MHz with a center frequency of 3.7 GHz. The simulation to fit the insertion losses at center frequency (0.28 dB) is carried out by means of an effective conductivity that takes into account the conductivity of the aluminum, the roughness of the metallic surfaces, the coupling and tuning screws, the coaxial probes and the contact between the body and the cover. This effective conductivity $\sigma_{eff}=2.8$ MS/m is lower than the theoretical conductivity of aluminum because of the above considerations.

Finally, Table 5 shows a comparison of the devised triple-mode filter with some prior-art designs. As can be observed, this work is the only one that implements a fully canonical narrow-band response.

## 5. Conclusions

The design of a triple-mode waveguide filter for application in the 5G band is reported. The filter implements a fully canonical response to improve the actual selectivity provided by previous designs by means of synthesizing the maximum possible number of transmission zeros, i.e., the order of the filter with a single cavity. Coaxial probes are the input–output ports to control and achieve the required level of the source–load coupling. In addition, microwave precision tuning screws allow for controlling the asynchronous resonating frequency of the three modes and the position of the transmission zeros.

Moreover, a proof-of-concept prototype has been manufactured and measured, showing an excellent agreement between the simulation and the experimental characterization.

## Figures and Tables

**Figure 1 sensors-25-00090-f001:**
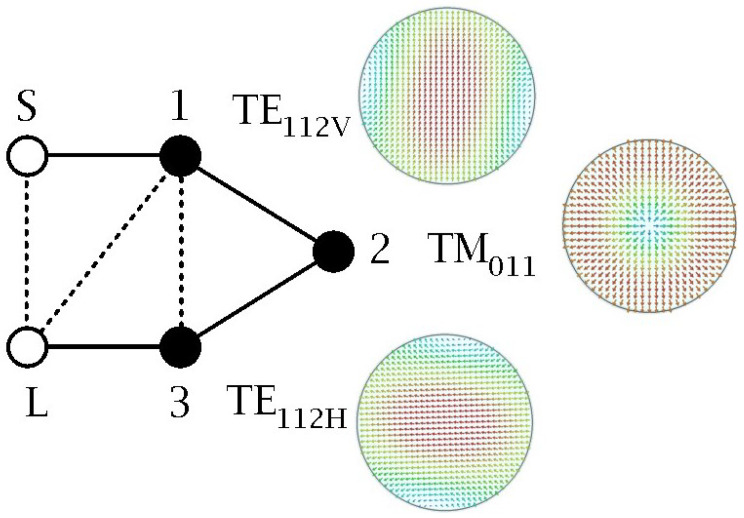
Folded *N* + 2 fully canonical network with *N* = 3: source (S) and load (L) terminals (circle), resonator nodes (solid circle), mainline couplings (solid line) and cross-couplings (dashed line). The first resonator corresponds to the $TE_{112V}$ mode, the second resonator to the $TM_{011}$ mode and the third resonator to the $TE_{112H}$ mode.

**Figure 2 sensors-25-00090-f002:**
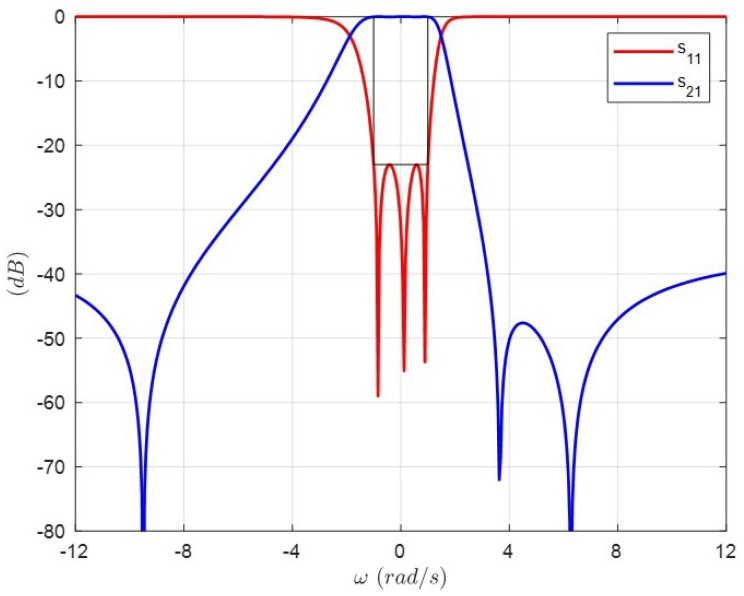
Low-pass prototype transmission and reflection characteristics of the (3-3) Chebyshev filter with three prescribed transmission zeros at $s_{1}=-j9.5$, $s_{2}=j3.65$ and $s_{3}=j6.27$. The three coupling matrices in Table 2 synthesize this response.

**Figure 3 sensors-25-00090-f003:**
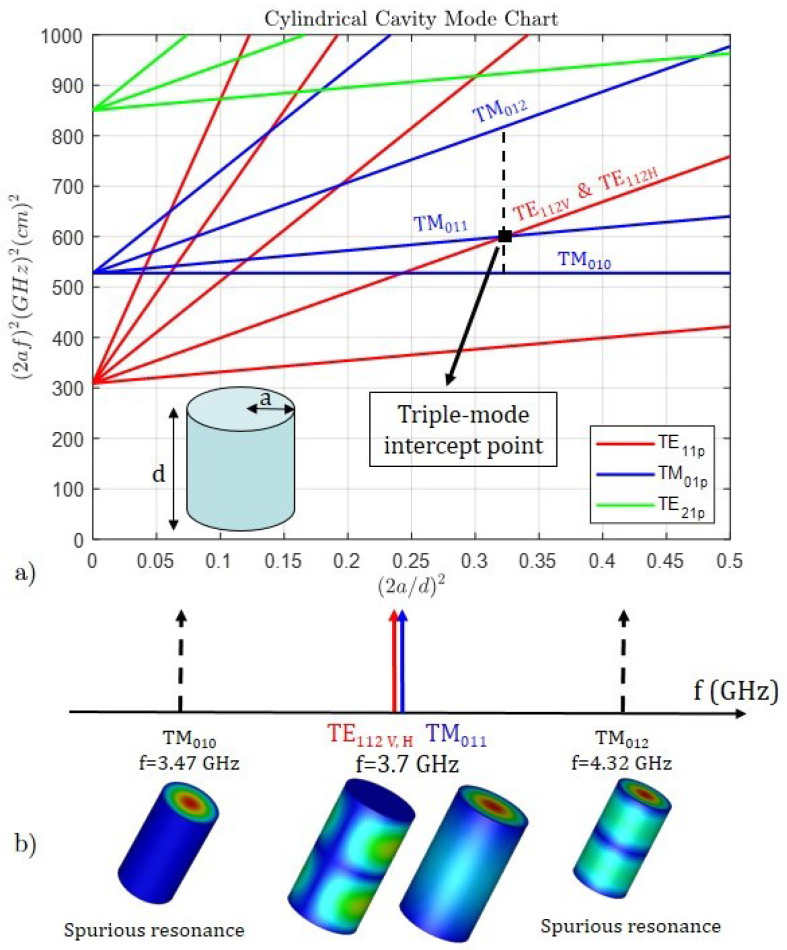
(**a**) Cylindrical cavity mode chart and the selected point, intersection of $TE_{112V,H}$ and $TM_{011}$ modes, for the design of the triple-mode filter: radius a = 33.11 mm and length d = 116.41 mm for resonant frequency f=3.7 GHz. (**b**) Free spurious window of 850 MHz; the resonant frequency of the lower spurious mode $TM_{010}$ is 3.47 GHz and the resonant frequency of the upper spurious mode $TM_{012}$ is 4.32 GHz.

**Figure 4 sensors-25-00090-f004:**
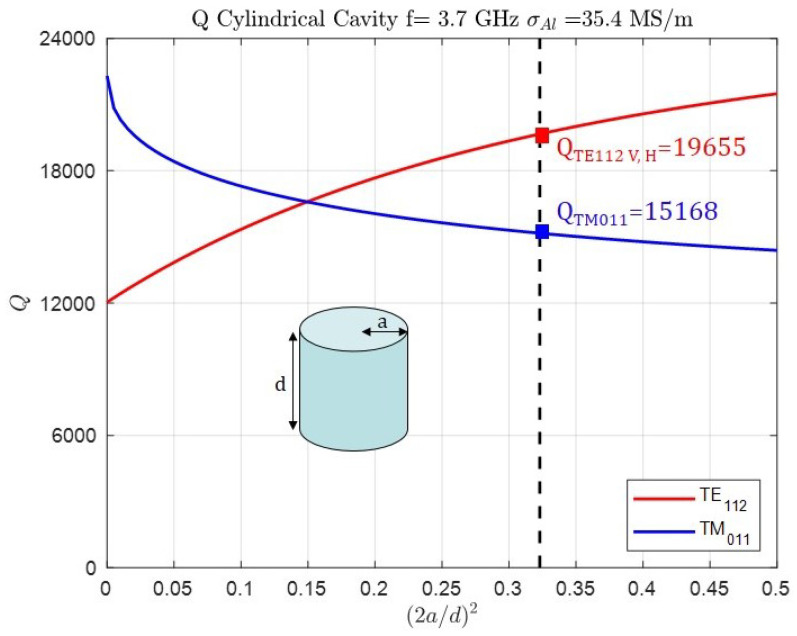
Q factor of the $TE_{112V}$ and $TE_{112H}$ modes and the $TM_{011}$ mode at frequency f=3.7 GHz, considering aluminum metallic walls ($\sigma_{Al}$ = 35.4 MS/m) and the abscissa design ${\left(2a/d\right)}^{2}$ = 0.323, radius a = 33.11 mm and length d = 116.41 mm.

**Figure 5 sensors-25-00090-f005:**
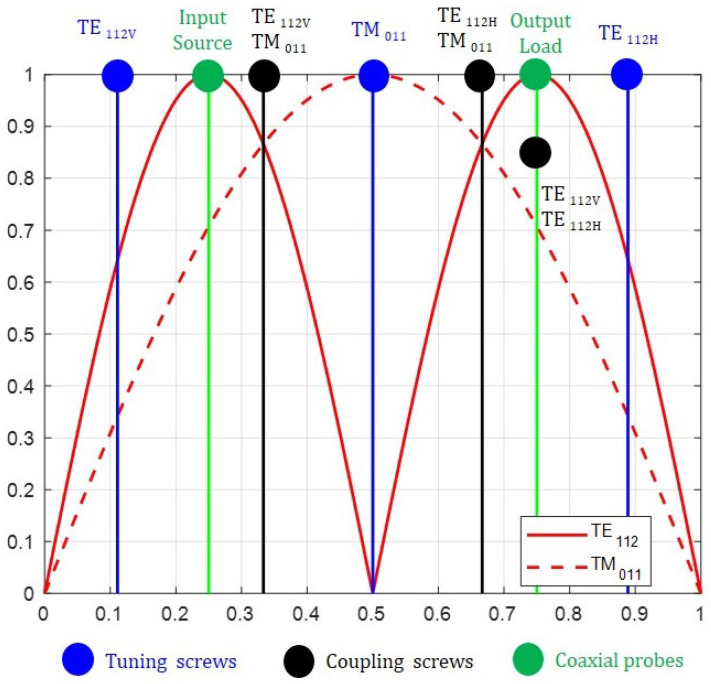
Magnitude of the transversal component of the electric field for the $TE_{112V,H}$ and $TM_{011}$ modes versus the length of the cylindrical cavity normalized to unity. The position of the tuning and coupling screws and the input–output coaxial probes.

**Figure 6 sensors-25-00090-f006:**
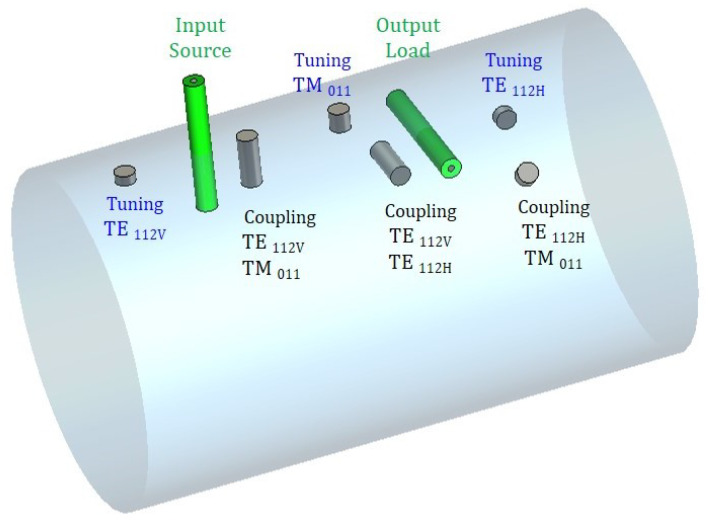
A 3D view of the cavity showing the tuning and coupling screws and the input–output coaxial probes.

**Figure 7 sensors-25-00090-f007:**
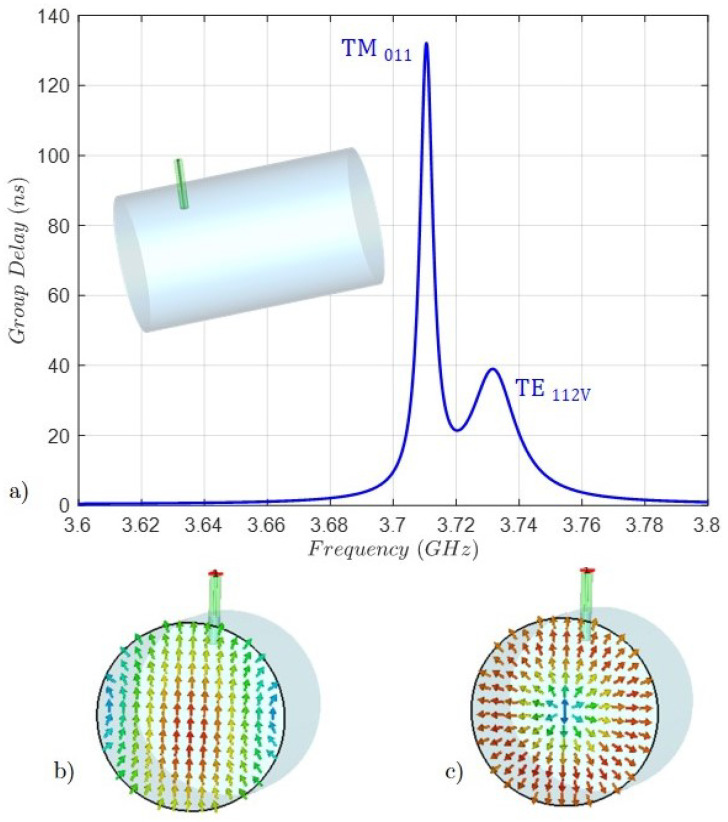
(**a**) Group delay of the input reflection coefficient, the $s_{11}$ parameter, and the 3D view of the cavity with the input coaxial probe. (**b**) The transversal component of the electric field of the $TE_{112V}$ mode at 3.73 GHz. (**c**) The transversal component of the electric field of the $TM_{011}$ mode at 3.71 GHz. Both frequencies are above the center frequency of 3.7 GHz, which will be decreased by the tuning screws.

**Figure 8 sensors-25-00090-f008:**
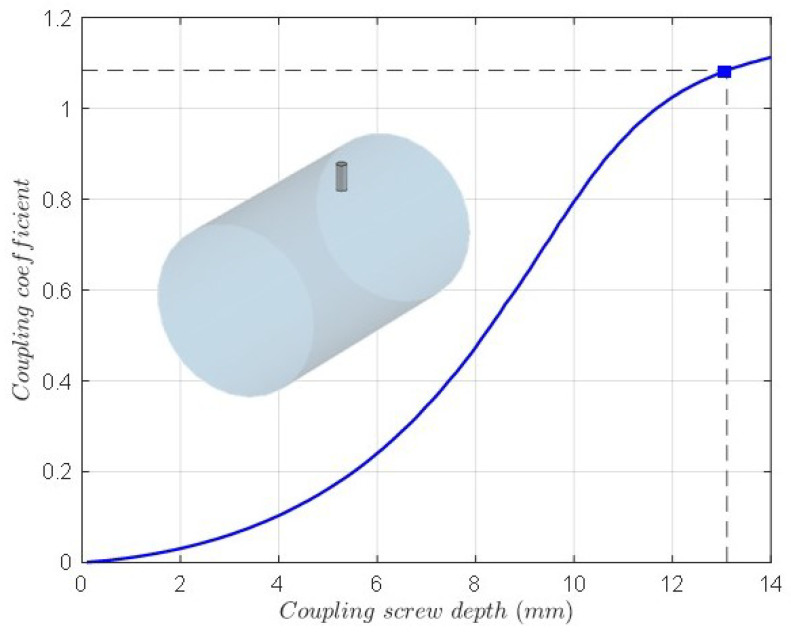
The coupling coefficient versus the depth of the screw for the coupling between the $TE_{112V}$-$TM_{011}$ modes and $TE_{112H}$-$TM_{011}$ modes, including the 3D model of the cavity and the coupling screw.

**Figure 9 sensors-25-00090-f009:**
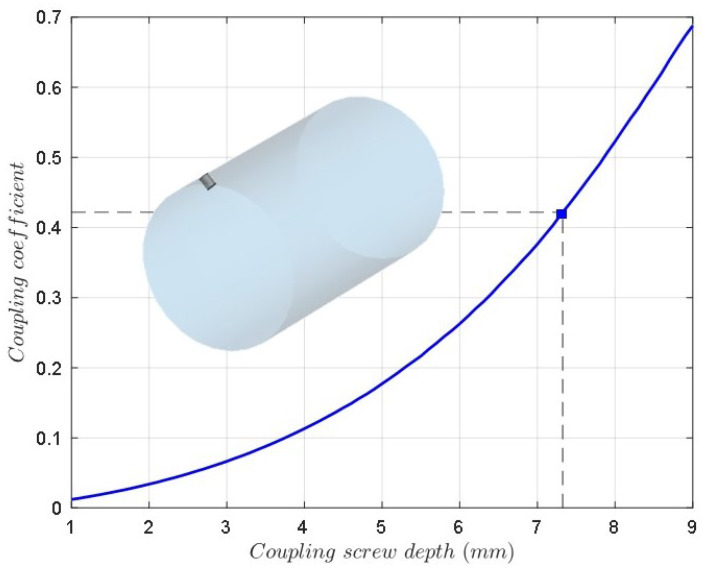
The coupling coefficient versus the depth of the screw for the coupling between the $TE_{112V}$-$TE_{112H}$ modes, including the 3D model of the cavity and the coupling screw at $45^{\circ }$.

**Figure 10 sensors-25-00090-f010:**
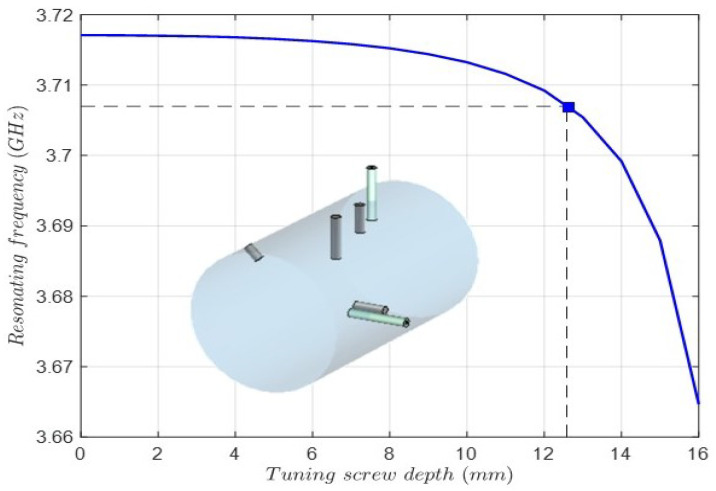
Cavity resonant frequency versus the tuning screw depth for the $TM_{011}$ mode. In the inset is the 3D model of the cavity, including the input–output coaxial probes and the coupling screws.

**Figure 11 sensors-25-00090-f011:**
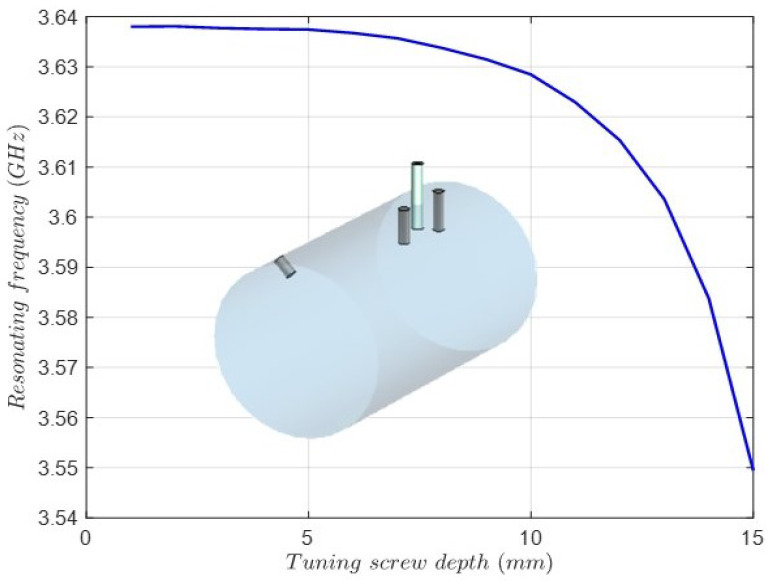
Cavity resonant frequency versus the tuning screw depth for the $TE_{112V}$ mode. In the inset is the 3D model view of the metal cavity and the tuning screw for the $TE_{112V}$ mode and the vertical coaxial probe and the coupling screws for the $TE_{112V}$-$TM_{011}$ modes and the $TE_{112V}$-$TE_{112H}$ modes.

**Figure 12 sensors-25-00090-f012:**
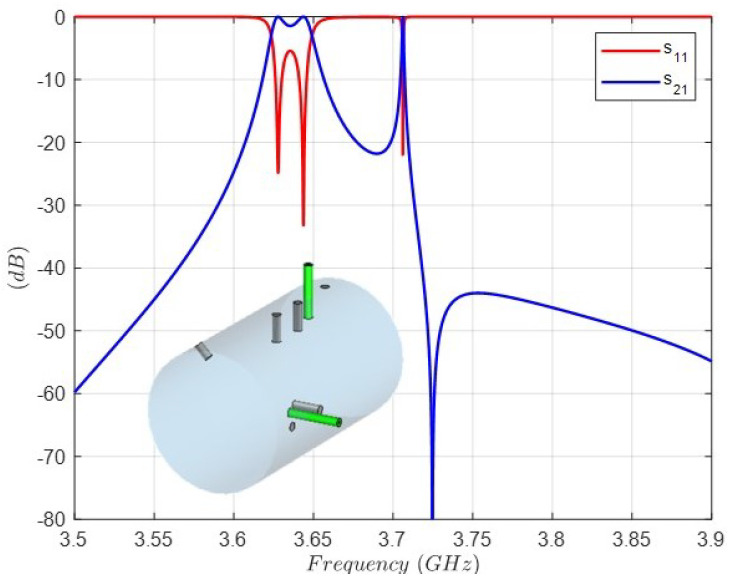
Full-wave simulation of the triple-mode filter with the geometry corresponding to the initial values collected in Table 3.

**Figure 13 sensors-25-00090-f013:**
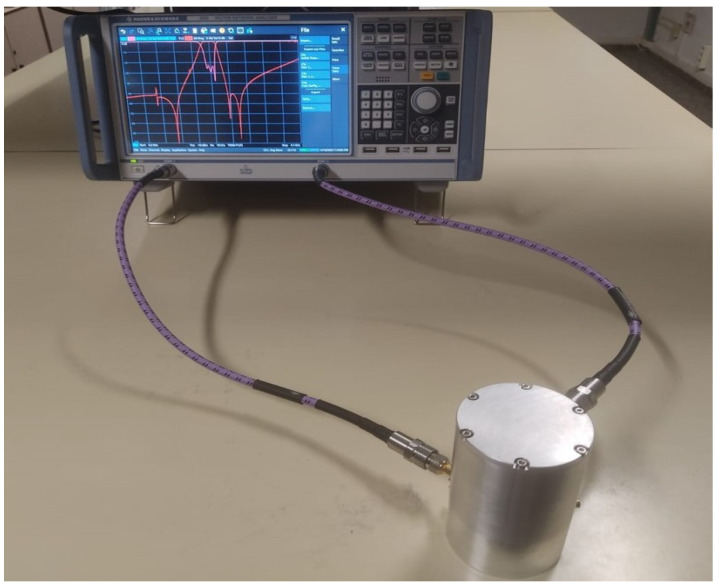
S-parameters of the simulation and measurement in the working bandwidth.

**Figure 14 sensors-25-00090-f014:**
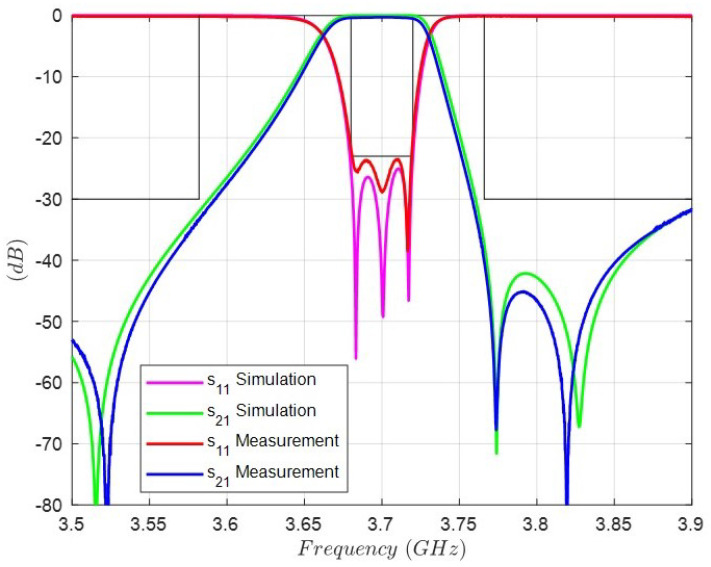
Comparison between the full-wave simulation and the measurement of the S-parameters of the triple-mode filter.

**Figure 15 sensors-25-00090-f015:**
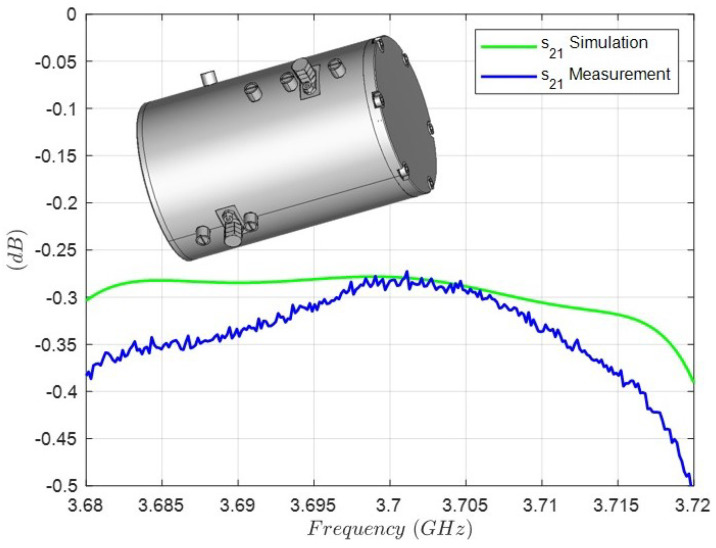
Comparison between the simulation and the measurement of the insertions losses of the triple-mode filter, considering an effective conductivity of $\sigma_{eff}=2.8$ MS/m.

**Table 1 sensors-25-00090-t001:** Triple-mode bandpass filter specifications.

Order N	3
Center frequency $f_{o}$	3.7 GHz
Bandwidth BW	40 MHz
Transmission zeros	$s_{1}=-j9.5\;,s_{2}=j3.65\;,s_{3}=j6.27$
Return loss	23 dB
Insertion loss	<0.5 dB
Input–output ports	SMA 50 Ohms coaxial

**Table 2 sensors-25-00090-t002:** Values for three different synthesized coupling matrices according to (1).

	Matrix 1	Matrix 2	Matrix 3
$m_{S1}$	1.1640	1.6326	139.6593
$m_{SL}$	0.0083	0.9835	−119.9802
$m_{11}$	0.1048	0.1006	−0.4009
$m_{12}$	1.0810	1.0810	1.0810
$m_{13}$	−0.4234	1.7672	−162.9893
$m_{1L}$	0.0018	−0.0025	0.2172
$m_{22}$	−0.3780	−0.3780	−0.3780
$m_{23}$	−1.0827	1.0827	−1.0827
$m_{33}$	0.1061	0.1061	0.1061
$m_{3L}$	1.1640	1.6326	139.6591

**Table 3 sensors-25-00090-t003:** Initial dimensions of the triple-mode filter.

Parameter	Value (mm)
Cavity radius	33.09
Cavity length	113
Coaxial probe length	10
Screw radius	2
$TE_{112V}$-$TM_{011}$ coupling screw depth	13.11
$TE_{112H}$-$TM_{011}$ coupling screw depth	13.11
$TE_{112V}$-$TE_{112H}$ coupling screw depth	7.32
$TM_{011}$ tuning screw depth	12.59
$TE_{112}$ tuning screw depth	0

**Table 4 sensors-25-00090-t004:** Final dimensions of the triple-mode filter.

Parameter	Value (mm)
Cavity radius	33.37
Cavity length	110.51
Coaxial probe length	12.54
Screw radius	2
$TE_{112V}$-$TM_{011}$ coupling screw depth	0.32
$TE_{112H}$-$TM_{011}$ coupling screw depth	0.28
$TE_{112V}$-$TE_{112H}$ coupling screw depth	2.49
$TM_{011}$ tuning screw depth	0.12
$TE_{112}$ tuning screw depth	0

**Table 5 sensors-25-00090-t005:** Comparison with some related prior-art triple-mode filters.

Characteristic	[17]	[20]	[21]	[23]	This Work
Number of resonant modes	3	3	3	3	3
Number of Tzs	1	3	2	2	3
Controllable Tzs	Yes	No	No	Yes	Yes
Type of intermode coupling	Coupling screw (totally tunable)	Metal disk and post (no tunable)	Metal cylinder (no tunable)	Metal disk and post (no tunable)	Coupling screw (totally tunable)
Type of external coupling	Rectangular waveguide	Coaxial SMA	Coaxial SMA	Coaxial SMA	Coaxial SMA
Center frequency CF	12 GHz	2.5 GHz	3.3 GHz	2.45 GHz	3.7 GHz
Fractional bandwidth	0.75%	40%	30%	3.4%	1.1%
Return losses	23 dB	17.5 dB	20 dB	20 dB	23 dB
Insertion losses at CF	0.43 dB	0.28 dB	0.5 dB	0.18 dB	0.28 dB
Unloaded Q (resonant modes)	N/A	$TM_{010}$— 2039 $TE^{+}$—1284 $TE^{-}$—2236	TM—4828 $TE^{+}$—4672 $TE^{-}$—4941	N/A	$TM_{011}$—19,655 $TE_{112V,H}$—15,168
Size Wavelength $\lambda_{c}$ at CF	Diameter: 0.89$\lambda_{c}$ Length: 1.92$\lambda_{c}$	0.46 × 0.19 × 0.12$\lambda_{c}$	0.5 × 0.5 × 0.56$\lambda_{c}$	0.03 × 0.03 × 0.05$\lambda_{c}$	Diameter: 0.82$\lambda_{c}$ Length: 1.36$\lambda_{c}$

## Data Availability

The data are contained within this article.

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
