# Peer review of "Fully Canonical Triple-Mode Filter with Source-Load Coupling for 5G Systems"

_sensors, 2024, doi:10.3390/s25010090_

Round 1

Reviewer 1 Report

Comments and Suggestions for Authors

Dear Authors

 This paper presents a fully canonical three-mode filter with source-load coupling in a circular waveguide. The coupling matrix is provided, and the size design of the relevant physical structure is analyzed. Both simulation results and test results are presented. To further improve the content, there are some concerns about the manuscript as following:

(1) Page 6 paragh 2, “Fig. 4 shows the Q factor of the three modes under analysis versus the aspect ratio ratio of the cylindrical cavity”, there are two words “ratio”.

(2) The paper is written in considerable detail, with a few transitional phrases that give it the impression of being a design report. It is recommended to revise these phrases to make them more consistent with academic papers.

(3) “Fig. 9 shows the coupling value (m13) versus the depth of the screw for coupling between TE112V-TE112H modes”. The term "coupling value", in the filter community, we generally refer to it as the "coupling coefficient." For the coupling coefficient in the case of asynchronously tuned, it is recommended to provide a specific calculation formula.

Author Response

Thank you very much for taking the time to review this manuscript. Please find the detailed responses below.

Comments:

(1) Page 6 paragh 2, “Fig. 4 shows the Q factor of the three modes under analysis versus the aspect ratio ratio of the cylindrical cavity”, there are two words “ratio”.

The typo has been corrected.

(2) The paper is written in considerable detail, with a few transitional phrases that give it the impression of being a design report. It is recommended to revise these phrases to make them more consistent with academic papers.

Following the comment, some phrases have been rewritten to be more consistent with academic papers.

(3) “Fig. 9 shows the coupling value (m13) versus the depth of the screw for coupling between TE112V-TE112H modes”. The term "coupling value", in the filter community, we generally refer to it as the "coupling coefficient." For the coupling coefficient in the case of asynchronously tuned, it is recommended to provide a specific calculation formula.

The term "coupling value" has been replaced by "coupling coefficient" in the text and in the axis of figures 8 and 9.

The coupling coefficient m13 between TE112V-TE112H modes is calculated by the formula (7.41) in reference [30]. The physical model with the screw at 45o degrees in figure 9 is used to find the two eigenvalues corresponding to the two modes electrically coupled. This process does not take into account that the filter is asynchronously tuned since the goal is to obtain the initial depth of the screw. In any case formula (7.52) for asynchronously tuned coupled resonator circuits with electric coupling is the same that formula (7.41) for synchronously tuned resonator, i.e., the relationship between both eigenvalues.

Reviewer 2 Report

Comments and Suggestions for Authors

Dear Authors

Thank you for sending an interesting article. Designing and creating such type filters is not easy. Of course, sometimes tuning them to the required parameters also causes a lot of problems. The article is written clearly and understandably.  

However, I have a few comments.  

  1. 1. Line 47. You wrote that cylindrical waveguides have low insertion loss. Are they really better than rectangular (square) waveguides in this respect? 

  2.  
  1. 2. Figure 8 shows screw depth to obtain coupling value of 1.08 (it is above 13mm) In figure 10 screw depth to obtain resonating frequency is shown. Why is the figure not showing frequency 3.7 but a bit higher, i.e. around 3.72? I have not found a clear explanation. Maybe it would be good to do it. 

  2.  
  1. 3. Line 146. It is written that the tuning of the three modes is independent. When tuning the filter, do you really have to follow a specific order of screw adjustments, or is it arbitrary? (Some information is on lines 256-258). In other words, 3 screws are used for tuning. Do you have to start tuning from a specific one, or is it arbitrary? 

  2.  
  1. 4. In the text You mention C band. Have you checked what is the range of the filter's center frequency and its bandwidth. As you know, 5G can use 100MHz bandwidth in C band. Your filter has a 3 dB bandwidth of about 70MHz.

  2.  
  1. 5. In line 248 you write about “the high sensitivity of the response to variant in elements in Filter Design 3D”. Do you also observe such high sensitivity in the S parameters? If so, it might be a good idea to add a figure with the results for one unadjusted screw. You can also make a parametric study. Of course, instead of figures and parametric study, the issue can be described in a few sentences. I leave the decision to You.

Author Response

Thank you very much for taking the time to review this manuscript. Please find the detailed responses below.

Comments:

1) Line 47. You wrote that cylindrical waveguides have low insertion loss. Are they really better than rectangular (square) waveguides in this respect? 

In reference [5], figure 5.06-1 shows the attenuation data for rectangular and circular waveguides. As can be observed, the circular waveguide is better than the rectangular in this respect.

2) Figure 8 shows screw depth to obtain coupling value of 1.08 (it is above 13mm) In figure 10 screw depth to obtain resonating frequency is shown. Why is the figure not showing frequency 3.7 but a bit higher, i.e. around 3.72? I have not found a clear explanation. Maybe it would be good to do it. 

The reason lies in the fact that the other screws introduced in the adjustment process will lower the resonance frequency. Therefore, a higher frequency of 3.72 GHz is used as starting point to have a larger setting range.

3) Line 146. It is written that the tuning of the three modes is independent. When tuning the filter, do you really have to follow a specific order of screw adjustments, or is it arbitrary? (Some information is on lines 256-258). In other words, 3 screws are used for tuning. Do you have to start tuning from a specific one, or is it arbitrary? 

The adjustment process is a bit tricky, since at the end, there is a mutual interaction between the coupling and the tuning screws. The process stars tuning the three different frequencies of the resonant modes since the filter is asynchronously tuned. Once the three reflection zeros are on the screen of the VNA the next step is to obtain the three transmission zeros increasing gradually the return loss level but always considering the coupling matrix measured in real time. The procedure requires a continuous feedback and patience.

4) In the text You mention C band. Have you checked what is the range of the filter's center frequency and its bandwidth. As you know, 5G can use 100MHz bandwidth in C band. Your filter has a 3 dB bandwidth of about 70MHz.

The specifications of the filter are collected in table 1. In this work the bandwidth is 40 MHz but is possible to extend to 100 MHz, as you state, without major setback.

5) In line 248 you write about “the high sensitivity of the response to variant in elements in Filter Design 3D”. Do you also observe such high sensitivity in the S parameters? If so, it might be a good idea to add a figure with the results for one unadjusted screw. You can also make a parametric study. Of course, instead of figures and parametric study, the issue can be described in a few sentences. I leave the decision to You.

   A new paragraph has been added in this respect (lines 258-262):

Note that the filter is extremely sensitive to any change in the position of the screws or probes, requiring great care in tuning since all the effects are simultaneously present in the physical cavity. In addition, the manufacturing tolerances must be compensated. It has to do with, in short, an adjustment process with eight elements, two probes, and six screws.